

# Modeling emotional contagion in the COVID-19 pandemic: a complex network approach

Yanchun Zhu[1], Wei Zhang[2] and Chenguang Li[3]

[1] Business School, Beijing Normal University, Beijing, China
[2] School of Information, Central University of Finance and Economics, Beijing, China
[3] School of Insurance, Central University of Finance and Economics, Beijing, China

## ABSTRACT

During public health crises, the investigation into the modes of public emotional contagion assumes paramount theoretical importance and has significant implications for refining epidemic strategies. Prior research predominantly emphasized the antecedents and aftermath of emotions, especially those of a negative nature. The interplay between positive and negative emotions, as well as their role in the propagation of emotional contagion, remains largely unexplored. In response to this gap, an emotional contagion model was developed, built upon the foundational model and enriched from a complex network standpoint by integrating a degradation rate index. Stability analyses of this model were subsequently conducted. Drawing inspiration from topological structural features, an enhanced model was introduced, anchored in complex network principles. This enhanced model was then experimentally assessed using Watts-Strogatz's small-world network, Barabási-Albert's scale-free network, and Sina Weibo network frameworks. Results revealed that the rate of infection predominantly dictates the velocity of emotional contagion. The incitement rate and purification rate determine the overarching direction of emotional contagion, whereas the degradation rate modulates the waning pace of emotions during intermediate and later stages. Furthermore, the immunity rate was observed to influence the proportion of each state at equilibrium. It was discerned that a greater number of initial emotional disseminators, combined with a larger initial contagion node degree, can amplify the emotion contagion rate across the social network, thus augmenting both the peak and overall influence of the contagion.

## INTRODUCTION

The global perturbation elicited by the COVID-19 pandemic has undeniably captured significant attention, influencing public emotional well-being and societal mental outlook (*Grima, Dalli Gonzi & Thalassinos, 2020*; *Okafor et al., 2022*; *Rahmadana, Loo & Aditia, 2022*; *Rudenstine et al., 2023*). With the virus demonstrating evolutionary tendencies through mutations, and a mounting uncertainty about the pandemic's trajectory, an accumulation of negative emotions, notably anxiety and anger, has been observed

Corresponding author
Chenguang Li, kddzw@163.com

among the populace (*Yousef et al., 2022*; *Zhao & Zhou, 2020*). Concurrently, preventative measures, encompassing mask-wearing, social distancing, enforced isolation, and remote working, though critical for containment, have inadvertently exacerbated the proliferation of negative emotions such as fear and resentment on social networks. Manifestations of these intensified sentiments are evident in rumor mongering, panic-induced actions, and regional stigmatization (*Dong et al., 2020*; *Okafor et al., 2022*; *Zhu et al., 2023*).

In the bid to refine public health emergency management, a profound understanding of emotional contagion laws and strategies to mitigate these adverse sentiments is deemed essential (*Cai et al., 2022*; *Zhang, Wang & Zhu, 2020*). Throughout the course of the COVID-19 pandemic, a predominance of studies have elucidated an augmented risk associated with negative emotional arousal, predisposing individuals to psychological afflictions such as depression and anxiety (*Giri & Maurya, 2021*; *Low et al., 2021*; *Waugh, 2020*). Yet, the symbiotic interaction between positive and negative emotions during pandemics and its consequent influence on emotional contagion has remained relatively understudied (*Basch, Corwin & Mohlman, 2021*; *Zhang, Wang & Zhu, 2020*).

Diverging from conventional research paradigms, which predominantly relied upon interviews, questionnaires, and psychological assessments, simulation modeling, anchored notably in the susceptible-infectious-susceptible (SIS) and susceptible-infectious-recovered/removed (SIR) epidemiological frameworks, offers an insightful lens to probe emotional contagion (*Hong et al., 2022*; *Wang et al., 2021*). Embracing this methodology not only facilitates the exploration of emotional contagion mechanisms, evolutionary patterns, and contagion scales (*Bakir, 2022*; *Iriany et al., 2023*; *Khare & Kaloni, 2022*; *Mahalingam & Pandraju, 2021*; *Widowati et al., 2022*; *Yang et al., 2022*; *Zhu et al., 2023*) but also transcends the boundaries set by traditional research. Albeit scholars have meticulously designed various simulation models employing theoretical constructs like machine learning, control theory, and artificial intelligence to decode the emotional infection-evolution mechanism, the majority ostensibly segment the population from a disease-contagion viewpoint, sidelining the dynamic interplay of emotions. Such models often posit that post-immunity individuals disengage from pertinent events, a supposition misaligned with real-world dynamics.

Incorporating the inherent emotional contagion attributes observed in netizens during the pandemic's ambient, and drawing upon classic infectious disease models enriched by complex network theory, an emotional contagion model delineating the dynamic evolution of emotional states has been proposed. Leveraging real-world scenarios, specifically the COVID-19 flare-up in Sanya, Hainan, in August 2022, the modulating role of critical parameters in the model concerning emotional contagion has been examined *via* simulation.

This research's contributions are multifaceted: (1) The degradation rate, resonating with the COVID-19 pandemic's nuances, has been integrated, segmenting emotion disseminators into positive and negative spectra, culminating in the construction of the SIpInRS model for netizen emotion contagion. (2) Simulation experiments have elucidated the intricate mechanisms by which infection, incitement, and purification rates

steer emotional contagion. (3) Adopting a complex network topology perspective, internodal contagion probability functions have been defined, factoring in mutual interplay between adjacent nodes, leading to enhancements in the SIpInRS model.

## RELATED RESEARCH

### Psychological aspects of emotion research

Central to both interpersonal and intrapersonal lives, emotions have been demonstrated to exert powerful effects, both advantageous and detrimental, on human functioning (*Nezlek & Kuppens, 2008*). With the onset of the COVID-19 pandemic, stress has been identified as a profound threat to public health, predisposing individuals to an array of negative emotional responses and thereby facilitating the development of mental health conditions such as depression and anxiety (*Liu et al., 2020*). Empirical evidence highlighting the intricate relationships between emotion and stress (*Prikhidko, Long & Wheaton, 2020*), depression (*Niu & Snyder, 2023*; *Whiston, Igou & Fortune, 2022*), and anxiety (*Muñoz Navarro et al., 2021*; *Wheaton, Prikhidko & Messner, 2021*) has been increasingly brought to the forefront. It is suggested by considerable research that the contagion of particularly negative emotions may culminate in outcomes such as the heightened prevalence of mental health symptoms, the employment of emotion regulation strategies conducive to resilience (*Low et al., 2021*; *Waugh, 2020*), the emotion diffusion effect (*Yu et al., 2022*), and the enhancement of psychological resilience (*Giri & Maurya, 2021*).

However, it was observed that a significant portion of the extant research emphasizes the Five-Factor Model of personality (*Kotov et al., 2010*) and predominantly employs psychological experiments (*Varma et al., 2023*), questionnaires (*Low et al., 2021*), and semi-structured interviews (*Srifuengfung et al., 2021*) to analyze emotional states. Consequently, less attention has been devoted to the interaction between positive and negative emotions and their respective contagion mechanisms (*Basch, Corwin & Mohlman, 2021*; *Zhang, Wang & Zhu, 2020*).

### Emotional infection models

The infectious disease model has traditionally served as an esteemed method for investigating the laws governing emotional contagion. In their exploration, *Hill et al. (2010)* categorized infection states into positive and negative, leading to the construction of the SISa emotional contagion model. A notable modification to the classical susceptible-infectious-susceptible (SIS) model, which included the spontaneous infection rate, was introduced under the presumption that emotional contagion could stem from factors unrelated to direct contacts. Significant findings were reported by *Fu et al. (2014)*, who combined the meta-cellular automata model with infectious disease theory, and by *Zhao et al. (2014)*, who advanced the SIRS model to study the dynamism of panic spread within subways. By integrating the OCEAN Big Five personality with infectious disease models, *Cao et al. (2017)* established a P-SIS model, which offered a more accurate representation of individual personalities during emergency evacuations. For the scenarios of crowd evacuation, other scholars have integrated traditional SIS and SIR models with diverse theories to propose various emotional contagion models such as the stochastic event-based

emotional contagion model (_Shi et al., 2021_; _Shang et al., 2023_) and personalized virtual and physical cyberspace-based emotional contagion model (_Hong et al., 2020_), dynamic multiple negative emotional susceptible-forwarding-immune model (_Yin et al., 2022_).

Yet, despite theoretical advancements, empirical evaluations of these models often lag, primarily due to the challenges in accurately measuring emotions within large groups (_Van Haeringen, Gerritsen & Hindriks, 2023_). Furthermore, scant attention has been paid to the impact of positive and negative emotion interaction on emotion contagion (_Zeng et al., 2022_). This gap was partially addressed by _Geng et al. (2023)_, who explored the influence of media interventions on online public opinion spread. However, their investigations remained devoid of a comprehensive examination of the laws governing emotional contagion amid positive and negative emotional interactions.

## Emotion contagion models based on complex networks

Given the limitations in contagion models, which often fail to mirror the genuine state of nodes within social networks, the adoption of complex network theory was initiated. For instance, _Yang et al. (2019)_ accounted for individual differences and network topologies in their research, proposing a rumor contagion ILSR model. In another study, _Xiong et al. (2018)_ posited that both spatial distance and time span significantly influence contagion within social networks, leading to the proposition of an emotion contagion model founded on multi-layered social networks. Another noteworthy contribution was made by _Wang et al. (2022)_ who introduced multilayer networks to study investor sentiment and stock return connectedness.

In conclusion, while extensive research has delved into the realms of emotional contagion, gaps still remain, particularly concerning the interaction between positive and negative emotions and their contagion mechanisms within complex networks. Addressing these gaps could provide a more holistic understanding of the dynamics of emotion contagion and its broader implications.

## Modeling and simulation analysis of emotional contagion among netizens amid the COVID-19 pandemic

### The formulation of a model depicting emotional contagion among netizens

In an attempt to capture the dynamism of emotional contagion among internet users during the pandemic, individuals exposed to varying information were categorized into positive and negative emotional states. Given the propensity for recurring emotional waves, a degradation rate was introduced. This rate implies that those previously immune bear a certain likelihood of becoming susceptible to emotions again, ensuring the model aligns closely with contagion dynamics observed during the pandemic.

Within this system, netizens are stratified into the ensuing states:

(1) Emotionally Susceptible (S): This category encompasses netizens yet to be exposed to pertinent information within the broader online community. While they maintain a specific initial emotional state, they display heightened vulnerability to emotional shifts upon encountering infected individuals.

(2) Positive Emotional Disseminators (Ip): Netizens within this category, after interacting with relevant content, demonstrate a capacity to process information rationally and

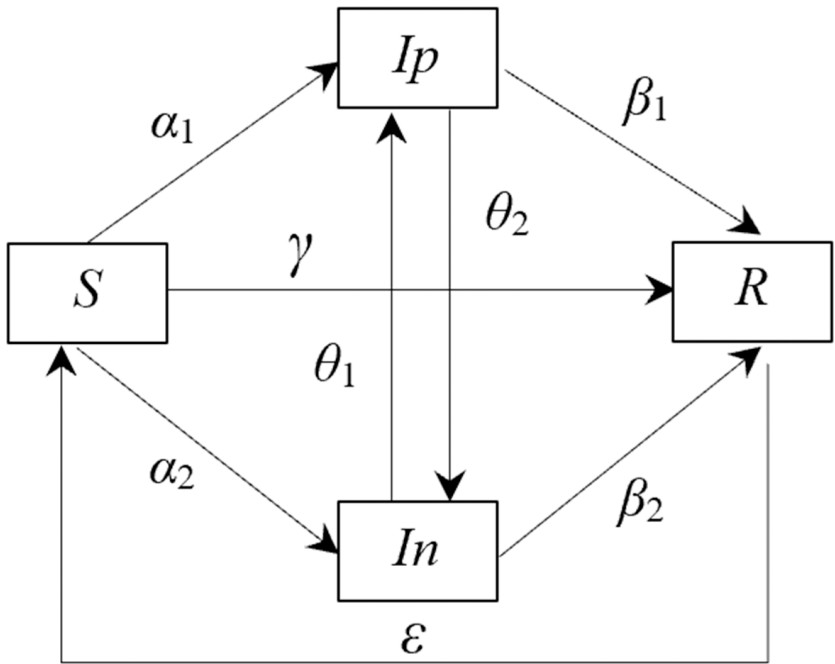

**Figure 1** **Relationship of various state transitions of netizen in the epidemic environment.** (A) Control group SIpInRS model. (B) Experimental group SIRS traditional model.

objectively. They disseminate optimism and positive emotions (defined as sentiments rooted in health, optimism, and constructive motivation) online. It was observed that such individuals can influence the emotional state of the susceptible and inadvertently drive the spread of negative sentiments.

(3) Negative Emotional Disseminators (In): This state embodies netizens with a predominant negative emotional disposition, susceptible to infection. These individuals perpetuate diverse negative emotions, such as panic, anxiety, and falsehoods, within the digital realm.

(4) Emotionally Immune (R): Netizens classified as emotionally immune remain unaffected by the prevalent emotional spectrum and refrain from transmitting associated sentiments.

Drawing from foundational assumptions and the infectious disease dynamics explored by *Geng et al. (2023)*, *Li et al. (2020)*; *Li, Liu & Li (2020)*, *Mao et al. (2019)*, and *Shen et al. (2022)*, a model elucidating the transitional relationships between these emotional states was formulated. Incorporating the interplay between positive and negative sentiments into the traditional SIRS framework, the SIpInRS transformation model, encapsulating the emotional shifts among netizens during the pandemic, was established, as depicted in Fig. 1.

As delineated in Fig. 1, $S(t)$, $Ip(t)$, $In(t)$, and $R(t)$ represent the ratios of netizens in each respective state to the overall netizen population at time $t$. It must be noted that within this system, $S(t)+In(t)+Ip(t)+R(t)=1$. The transformations inherent to each state are elucidated as follows:

(1) Positive Infection Rate $\alpha_1$: This rate depicts the probability by which an emotionally susceptible individual is influenced by positive emotions, transitioning into a positive emotion disseminator.

(2) Negative Infection Rate $\alpha_2$: Analogously, this rate signifies the likelihood that an emotionally susceptible individual, under the sway of negative emotions, becomes a negative emotion disseminator.

(3) Purification Rate $\theta_1$: This rate is used to represent the probability of a transition from a negative emotion disseminator to a positive emotion disseminator.

(4) Incitement Rate $\theta_2$: This value quantifies the likelihood that a positive emotion disseminator, upon exposure to negative emotional stimuli, transitions to a negative emotion disseminator.

(5) Direct Immunization Rate $\gamma$: Here, the probability is captured wherein an emotionally susceptible individual, displaying disinterest in the prevailing information, directly transitions to an emotionally immune state.

(6) Immunization Rates ($\beta_1$, $\beta_2$): These rates quantify the transition probabilities from both positive and negative emotional disseminators to an emotionally immune state.

(7) Degeneration Rate $\varepsilon$: This rate is representative of the likelihood that an emotionally immune individual reverts to an emotionally susceptible state after a certain period, possibly due to external factors such as environmental changes or memory decay.

The state transfer function encapsulating the transitions among the various netizen types, as proposed in this model, is grounded in the structure depicted in Fig. 1, hereby referred to as Model I.

Within this framework, $\alpha_1$ and $\alpha_2$ are understood to signify the likelihood of an emotionally susceptible individual's infection post-exposure to either a positive or negative emotional disseminator. $\gamma$ articulates the probability wherein a susceptible individual transitions directly to immunity post information exposure. $\theta_1$ and $\theta_2$, respectively, represent the likelihoods of a negative emotion disseminator converting due to positive influence (possibly governmental guidance) and a positive emotion disseminator succumbing to negative emotional provocation. Finally, $\beta_1$ and $\beta_2$ expound upon the transition probabilities for positive and negative disseminators to achieve emotional immunity, while $\varepsilon$ elaborates upon the propensity for immune individuals to re-enter susceptibility due to various external influences.

## Analysis of equilibrium points and stability within the model

Within the SIRS contagion framework, both zero and nonzero contagion equilibrium points have been identified (*Prodanov, 2021*). The eventual attainment of a steady state, whether it gravitates towards a zero or nonzero equilibrium, is predominantly contingent upon the contagion threshold, denoted as $R_0$. This threshold $R_0$ can be interpreted as the number of subsequent generations to which an infected individual can transmit within a singular time unit. It has been noted that should $R_0$ be less than or equal to 1, a zero-equilibrium state is ultimately approached. Under such circumstances, infected individuals within the netizen populace will be eradicated, resulting in a system exclusively comprising susceptible and immune individuals. Conversely, when $R_0$ exceeds 1, a nonzero equilibrium state

becomes the system's fate, encompassing susceptible, infected, and immune individuals concurrently.

Building upon prior definitions, the relationship $S(t)+In(t)+Ip(t)+R(t)=1$ is acknowledged. By setting the left sides of Eqs. (1)–(4) from Model I to zero, Model II is derived.

In the specified contagion model, parameters such as $\alpha_1, \alpha_2, \theta_1, \theta_2, \beta_1$ and $\beta_2$ are treated as constants. Consequently, with the relation $In(t)+Ip(t)=1$, both positive infection rate $\alpha_1$ and negative infection rate $\alpha_2$ are consolidated under a unified infection rate, $\alpha$. Similarly, both the positive immunity rate $\beta_1$ and negative immunity rate $\beta_2$ are uniformly denoted as $\beta$, leading to the derivation of Model III.

From the extrapolations of Model III, equilibria related to $I(t)$, are discerned. Depending on whether the equilibrium point is 0, the presence of an equilibrium point $P^*\left(\frac{\beta}{\alpha}, \frac{\varepsilon\alpha-\varepsilon\beta-\gamma\beta}{\alpha(\varepsilon+\beta)}\right)$, denoted as $P_0\left(\frac{\varepsilon}{\gamma+\varepsilon}, 0\right)$, where the number of infected individuals exceeds zero is determined. Hence, the contagion threshold can be ascertained by $R_0 = \frac{\varepsilon\alpha}{\varepsilon\beta+\gamma\beta}$. Should $R_0$ remain at or below 1, the system gravitates towards the equilibrium point $P_0$. In cases where $R_0$ surpasses 1, $P^*$ remains the sole equilibrium point. An in-depth examination into the stability of these equilibrium points ensues.

First, with $X=-\alpha S(t)I(t)-\gamma S(t)+\varepsilon(1-S(t)-I(t))$ and $Y=\alpha S(t)I(t)-\beta I(t)$, the matrix is obtained as follows.

$$\begin{bmatrix} \dfrac{\partial X}{\partial S}=-\alpha I(t)-\gamma-\varepsilon & \dfrac{\partial X}{\partial I}=-\alpha S(t)-\varepsilon \\ \dfrac{\partial Y}{\partial S}=\alpha I(t) & \dfrac{\partial Y}{\partial I}=\alpha S(t)-\beta \end{bmatrix} \tag{1}$$

Therefore, for the zero-equilibrium point under $P_0$, its Jacobian matrix $J$ is given by

$$\begin{bmatrix} -\gamma-\varepsilon & \dfrac{-\varepsilon\alpha}{\gamma+\varepsilon}-\varepsilon \\ 0 & \dfrac{\varepsilon\alpha}{\gamma+\varepsilon}-\beta \end{bmatrix} \tag{2}$$

Finding the eigenvalues of this matrix gives the matrix characteristic equation as $(\lambda+\gamma+\varepsilon)\left(\lambda-\frac{\alpha\varepsilon}{\gamma+\varepsilon}+\beta\right)$ because under the zero-equilibrium point $R_0 \leq 1$, Therefore, both eigenvalues are less than 0. According to the Routh-Huriwitz stability criterion, it is locally stable at zero equilibrium (*Mao et al., 2019*).

For the nonzero equilibrium point $P^*$, its Jacobian matrix $J$ is given by

$$\begin{bmatrix} -\dfrac{\varepsilon\alpha-\varepsilon\beta-\gamma\beta}{(\varepsilon+\beta)}-\gamma-\varepsilon & -\beta-\varepsilon \\ \dfrac{\varepsilon\alpha-\varepsilon\beta-\gamma\beta}{\varepsilon+\beta} & 0 \end{bmatrix} \tag{3}$$

The characteristic equation for this matrix is given by $\lambda^2+b\lambda+c=0$, where $c=(\beta+\varepsilon)\left(\frac{\varepsilon\alpha-\varepsilon\beta-\gamma\beta}{\varepsilon+\beta}\right)$, $b=\frac{\varepsilon\alpha-\varepsilon\beta-\gamma\beta}{(\varepsilon+\beta)}+\gamma+\varepsilon$, because at this equilibrium point, $R_0 \geq 1$.

Consequently, under these conditions, infected individuals persist over extended periods, culminating in a system where susceptible, infected, and immune individuals coexist in harmony within the network.

**Table 1  Parametric dispositions (SIpInRS vis-à-vis canonical model).**

| | S, Ip, In | $\alpha_1$ | $\alpha_2$ | $\beta_1$ | $\beta_2$ | $\gamma$ | $\theta_1$ | $\theta_2$ | $\varepsilon$ |
|---|---|---|---|---|---|---|---|---|---|
| 1. Reference group | 0.96, 0.02,0.02 | 0.45 | 0.40 | 0.06 | 0.04 | 0.1 | 0.05 | 0.02 | 0.05 |
| | S, I | | $\alpha$ | | $\beta$ | | | $\varepsilon$ | |
| 2. Experimental group SIRS | 0.96, 0.04 | | 0.45 | | 0.06 | | | 0.05 | |

In the realm of emotional contagion dynamics, the contagion threshold, denoted as $R_0$, emerges as a pivotal determinant of an affliction's potential to reach epidemic proportions. Empirical analyses consistently suggest that a diminished contagion threshold is indicative of superior overall control of the emotional contagion (*Al-Azzawi, 2012*; *Zhao et al., 2022*).

Upon examination of the deduced $R_0$ equation, it becomes evident that the contagion threshold within this epidemic emotional contagion framework is molded by the interplay of several parameters. These include the reduction of the negative emotional infection rate, symbolized as $\alpha$, augmentation of the direct immunity rate of susceptible individuals, represented by $\gamma$, and the modulation of the infected individuals' immunization rate, $\beta$. These modifications serve to effectively rein in the corresponding contagion threshold. Additionally, a decrement in the degradation rate, $\varepsilon$, alongside an extension in the effective duration of immunization for immunized entities, can further taper the contagion threshold. Hence, in the face of emergent situations, governmental interventions can aim to both curtail the primary sources of infection and rectify prevailing misinformation, thus attenuating the negative infection propensity amongst the general netizen populace. Such measures can invariably foster the metamorphosis of negatively influenced netizens into their positively charged counterparts. In parallel, harnessing the power of social media platforms to steer public sentiment in affirmative directions could expedite the transformation of infected individuals into immune ones, ultimately ensuring a balanced emotional landscape across digital spaces.

## Simulation analysis of digital emotional propagation models

Drawing upon methodologies delineated by *Li et al. (2020)* and *Zeng & Zhu (2019)*, parameters were set as follows: emotionally susceptible individual, $S = 0.96$; positive emotion disseminator, $Ip = 0.02$; negative emotion disseminator, $In = 0.02$; positive infection rate, $\alpha_1 = 0.45$; negative infection rate, $\alpha_2 = 0.4$; purification rate, $\theta_1 = 0.05$; incitement rate, $\theta_2 = 0.02$; immunization rate for positive disseminators, $\beta_1 = 0.06$; immunization rate for negative disseminators, $\beta_2 2 = 0.04$; direct immunization rate, $\gamma = 0.1$; and degeneration rate, $\varepsilon = 0.05$ (as tabulated in Table 1). In this segment, the SIpInRS model incorporating the aforementioned parameters serves as the reference model for ensuing simulation juxtapositions.

(1) Juxtaposition of the SIpInRS model and the canonical model

In relation to the canonical SIRS model, the SIpInRS paradigm, as illustrated in Fig. 2, not only segments disseminators based on emotional polarity but also accommodates the interplay between affirmative and negative emotions. The conventional approach amalgamates infected individuals into a singular cluster, obfuscating distinctions between the trajectories of positive and negative emotional transmissions. This homogenization

Zhu et al. (2023), *PeerJ Comput. Sci.*, DOI 10.7717/peerj-cs.1693

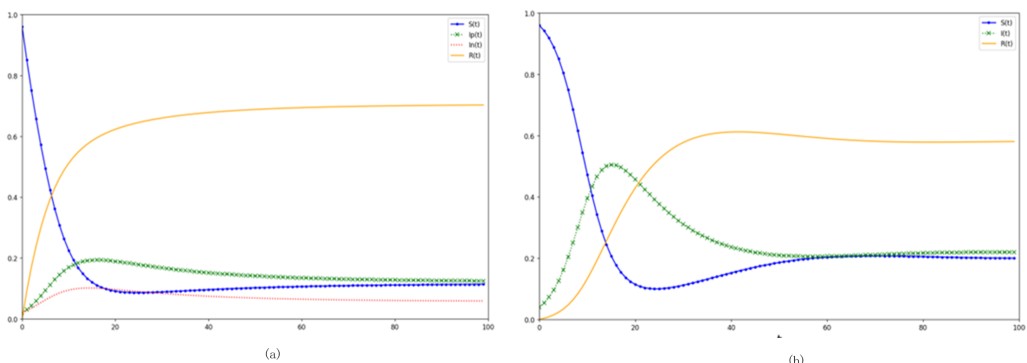

**Figure 2** **Dissection of the SIpInRS model in parallel with the canonical model.** (A) Control group SIp-InRS model. (B) Experimental group SIRS conventional model.

**Table 2** **Parametric alignments (pertaining to the influence of initial contagion ratios on digital sentiment progressions).**

|  | S, Ip, In | $\alpha_1$ | $\alpha_2$ | $\beta_1$ | $\beta_2$ | $\gamma$ | $\theta_1$ | $\theta_2$ | $\varepsilon$ |
|---|---|---|---|---|---|---|---|---|---|
| 1. Reference group | 0.96, 0.02, 0.02 | 0.45 | 0.40 | 0.06 | 0.04 | 0.1 | 0.05 | 0.02 | 0.05 |
| 2. Experimental group 1 | 0.90, 0.08, 0.02 | 0.45 | 0.40 | 0.06 | 0.04 | 0.1 | 0.05 | 0.02 | 0.05 |
| 3. Experimental group 2 | 0.90, 0.02, 0.08 | 0.45 | 0.40 | 0.06 | 0.04 | 0.1 | 0.05 | 0.02 | 0.05 |

complicates evaluations concerning the sway of these emotional polarities over variables like the incitement rate and purification rate. Furthermore, real-world scenarios often present netizens who, upon encountering pertinent information, transition directly from susceptibility to immunity, devoid of an interim infectious phase. Contrastingly, the canonical model postulates an inevitable progression from susceptibility through infection to immunity. This simplification results in an amplified peak of infectious individuals, concurrently stymieing the surge of immune proportions. Ergo, the augmented SIpInRS paradigm more accurately mirrors genuine contagion dynamics.

(2) Impacts of preliminary contagion proportions on digital sentiment evolution

Parameters were set as follows: positive infection rate $\alpha_1 = 0.45$, negative infection rate $\alpha_2 = 0.4$, purification rate $\theta_1 = 0.05$, incitement rate $\theta_2 = 0.02$, immunization rate $\beta_1 = 0.06$, immunization rate $\beta_2 = 0.04$, direct immunization rate $\gamma = 0.1$. degeneration rate $\varepsilon = 0.05$. For reference group, emotionally susceptible person $S = 0.96$, positive emotional disseminator $Ip = 0.02$, negative emotion disseminator $In = 0.02$; for experimental group1 and group2, $S = 0.90$, $Ip = 0.08$, $In = 0.02$; $S = 0.90$, $Ip = 0.02$, $In = 0.08$ respectively (in Table 2).

In juxtaposition with the reference model in Figs. 2A, 3A delineates the outcomes of augmenting the proportion of initial negative emotion propagators. An accelerated rise of negative emotional individuals within the entire digital ecosystem becomes discernible. The apex of negative emotion propagation is not only more pronounced but also noticeably shifted leftward. Moreover, during both the outbreak's inception and subsequent phases, the proportion of negative emotional individuals consistently

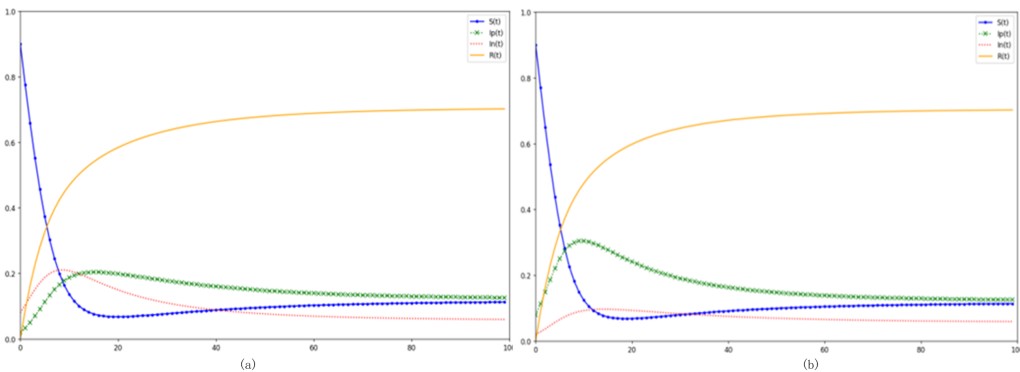

**Figure 3  Analysis of preliminary contagion ratios' influence on digital emotion evolution.** (A) Increase in the proportion of initial negative netizens. (B) Increase the proportion of initial positive netizens.

**Table 3  Parametric depictions (examining the emotional transmission rate's impact on digital sentiment evolution).**

|  | S, Ip, In | $\alpha_1$ | $\alpha_2$ | $\beta_1$ | $\beta_2$ | $\gamma$ | $\theta_1$ | $\theta_2$ | $\varepsilon$ |
|---|---|---|---|---|---|---|---|---|---|
| 1. Reference group | 0.96, 0.02, 0.02 | 0.45 | 0.40 | 0.06 | 0.04 | 0.1 | 0.05 | 0.02 | 0.05 |
| 2. Experimental group 1 | 0.96, 0.02, 0.02 | 0.45 | 0.80 | 0.06 | 0.04 | 0.1 | 0.05 | 0.02 | 0.05 |
| 3. Experimental group 2 | 0.96, 0.02, 0.02 | 0.80 | 0.40 | 0.06 | 0.04 | 0.1 | 0.05 | 0.02 | 0.05 |

eclipses their positive counterparts. Alternatively, in Fig. 3B, elevating the proportion of initial positive emotion propagators results in a more pronounced dominance of positive emotions at the network's onset. This dominance effectively steers susceptible individuals towards positivity, corroborating literature findings (*Li, Liu & Li, 2020*).

The simulations underscore that, during the early phases of emotional escalation in emergent scenarios, a limited cluster, encompassing primary stakeholders and a select group of affiliates, serve as the inaugural emotion propagators. However, when this nucleus of initial propagators expands, the incident's trajectory alters precipitously, culminating in an amplified emotional zenith. This intensification, in turn, resonates within public forums, culminating in an "echo chamber" phenomenon and fostering emotional symbiosis. Consequently, authoritative bodies ought to diligently monitor the proportion of primary emotional disseminators during crises, proactively implementing strategies to mitigate adverse emotional proliferations, thereby curtailing potential collateral damages.

(3) Influence of emotional transmission rates on digital emotion proliferation

Parameters were set as follows: emotionally susceptible person $S = 0.96$, positive emotional disseminator $Ip = 0.02$, negative emotion disseminator $In = 0.02$, purification rate $\theta_1 = 0.05$, incitement rate $\theta_2 = 0.02$, immunization rate $\beta_1 = 0.06$, immunization rate $\beta_2 = 0.04$, direct immunization rate $\gamma = 0.1$, degeneration rate $\varepsilon = 0.05$. For reference group, positive infection rate $\alpha_1 = 0.45$, negative infection rate $\alpha_2 = 0.4$,; for experimental group1 $\alpha_1 = 0.45$, $\alpha_2 = 0.8$, and group2, $\alpha_1 = 0.8$, $\alpha_2 = 0.4$ respectively (in Table 3).

Figures 4A and 4B elucidate variances ensuing from amplifications in positive transmission rates $\alpha_1$ and negative transmission rates $\alpha_2$, respectively. Observations

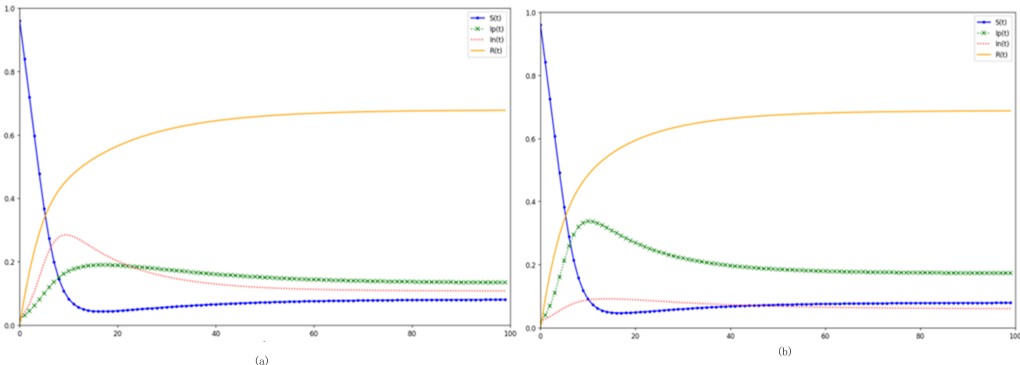

**Figure 4** **Analyzing the impacts of emotional transmission rates on emotion propagation.** (A) Increasing the rate of negative infection. (B) Increasing the rate of active infection.

from Fig. 4A indicate that a surging negative transmission rate precipitates an elevated peak in the $In(t)$ curve, manifesting earlier. Such a trend implies a heightened proclivity for netizens to succumb to negative emotions, thereby transitioning into negative emotion propagators during an event's nascent stage, subsequently inhibiting positive emotion diffusion. Drawing parallels with insights from *Tian, Sun & Zhang (2019)*, it's inferred that an elevated likelihood of negative transmission expedites the metamorphosis of susceptible individuals into negative emotional propagators, thereby precipitating a swift decline in the residual susceptible cohort. Analogous consequences, akin to those observed post-enhancement of primary negative propagators, manifest under this paradigm. However, a discernible deviation emerges in that transmission probabilities impinge upon the demographic distribution across varied states upon attaining a stable equilibrium.

The data underscores that the transmission rate $\alpha_2$ plays a pivotal role. It not only hastens the emotion dissemination speed within the digital realm and magnifies the upheaval experienced during an event's outbreak, but also precipitates swift shifts in the demographics of those afflicted with a particular emotion. This, in turn, mutes the counterbalancing influence of the opposing emotion. In tangible scenarios, a swift spread of negative emotions during the initial dissemination phase can inadvertently tip the balance on digital platforms towards negative sentiments, amplifying their proliferation. As the contagion phase matures, and in a setting where such emotional outbreaks are perceived as routine, neglecting to curtail the negative emotion transmission rate may culminate in a higher likelihood of immune individuals reverting to susceptibility and subsequently evolving into negative emotion propagators. This trend intensifies the overarching negativity within the digital ecosystem. From a sentiment management perspective, authoritative entities and social media platforms ought to offer nuanced guidance to netizens, facilitating a more balanced evaluation of emergent situations. By bolstering discernment capabilities and championing positivity, they can stymie the rampant spread of digital negativity.

(4) Dissecting the impacts of purification and incitement rates on digital emotion dissemination

**Table 4 Parametric configurations (exploring the emotional transmission rate's impact on digital sentiment evolution).**

| | $S, Ip, In$ | $\alpha_1$ | $\alpha_2$ | $\beta_1$ | $\beta_2$ | $\gamma$ | $\theta_1$ | $\theta_2$ | $\varepsilon$ |
|---|---|---|---|---|---|---|---|---|---|
| 1. Reference group | 0.96, 0.02, 0.02 | 0.45 | 0.40 | 0.06 | 0.04 | 0.1 | 0.05 | 0.02 | 0.05 |
| 2. Experimental group 1 | 0.96, 0.02, 0.02 | 0.45 | 0.40 | 0.06 | 0.04 | 0.1 | 0.1 | 0.02 | 0.05 |
| 3. Experimental group 2 | 0.96, 0.02, 0.02 | 0.45 | 0.40 | 0.06 | 0.04 | 0.1 | 0.05 | 0.05 | 0.05 |

Parameters were set as follows: emotionally susceptible person $S = 0.96$, positive emotional disseminator $Ip = 0.02$, negative emotion disseminator $In = 0.02$, positive infection rate $\alpha_1 = 0.45$, negative infection rate $\alpha_2 = 0.4$, immunization rate $\beta_1 = 0.06$, immunization rate $\beta_2 = 0.04$, direct immunization rate $\gamma = 0.1$, degeneration rate $\varepsilon = 0.05$. For reference group, purification rate $\theta_1 = 0.05$, incitement rate $\theta_2 = 0.02$; for experimental group1 $\theta_1 = 0.1$, $\alpha_2 = 0.02$, and group2, $\theta_1 = 0.05$, $\theta_2 = 0.05$ (in Table 4).

Figures 5A and 5B respectively modify the purification and incitement rates within the communication model. An augmented purification rate signals a higher propensity for a transition from negative to positive emotion propagators, engendering a more sizable positive sentiment cohort within the digital network. Utilizing Fig. 5B as a reference point—where the incitement rate is raised to $\theta_2 = 0.05$-a juxtaposition with the baseline model reveals that although the trajectories of susceptible $S(t)$ and immune $R(t)$ remain largely unchanged with an increased incitement rate, the $In(t)$ trajectory, indicative of negative sentiment propagators, is noticeably loftier, with its apex surpassing that of the control group. Conversely, the $Ip(t)$ trajectory is substantially muted. Collating this with the scholarly insights from (Shen et al., 2022), it becomes evident that a heightened incitement rate predominantly steers digital sentiment towards negativity, concurrently curtailing the ascendance of positive sentiment propagators. In contrast, enhancing the network's positive sentiment purification rate can markedly uplift the overarching digital sentiment. Both the incitement and purification rates impart distinct influences upon the eventual stabilized state. For instance, in the control configuration, the count of terminal positive sentiment propagators is roughly double that of their negative counterparts. However, post-amplification of the incitement rate, as observed in Fig. 5B, the terminal counts of both positive and negative sentiment propagators converge, nearly equalizing by the end.

(5) The dynamics of degradation rate on online emotional propagation

Parameters were set as follows: emotionally susceptible person $S = 0.96$, positive emotional disseminator $Ip = 0.02$, negative emotion disseminator $In = 0.02$, positive infection rate $\alpha_1 = 0.45$, negative infection rate $\alpha_2 = 0.4$, immunization rate $\beta_1 = 0.06$, immunization rate $\beta_2 = 0.04$, direct immunization rate $\gamma = 0.1$. For reference group, degeneration rate $\varepsilon = 0.05$; for experimental group1 $\varepsilon = 0.08$, and group2, $\varepsilon = 0.03$ (in Table 5).

In Fig. 6A, the degradation rate, representing the tendency of immune individuals to revert back to susceptibility-whether due to evolving external factors or inherent forgetfulness-is augmented. Meanwhile, Fig. 6B delineates the effects of a diminished degradation rate. When we observe the heightened degradation rate of Fig. 6A and juxtapose it with our baseline, a conspicuous pattern emerges: post-apex, the trajectories

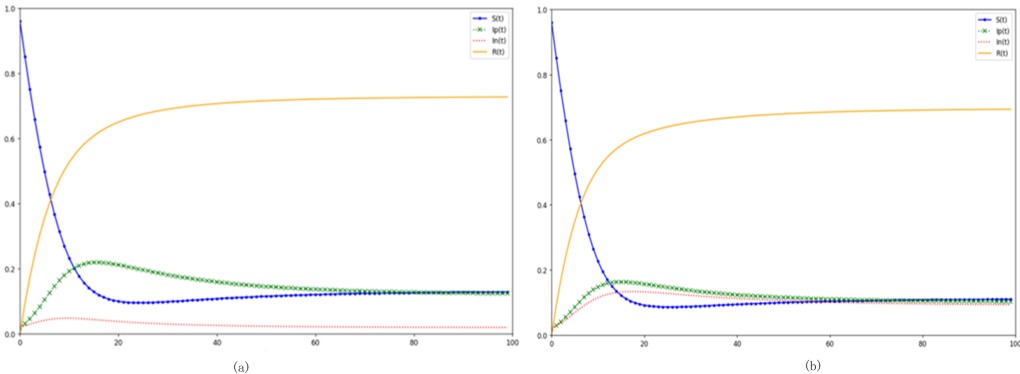

**Figure 5 Investigating the impacts of purification and agitation rates on digital emotion propagation.**
(A) Increased purification rate. (B) Increased incitement rate.

**Table 5 Configurational parameters (exploring the degradation rate's impact).**

|  | $S, Ip, In$ | $\alpha_1$ | $\alpha_2$ | $\beta_1$ | $\beta_2$ | $\gamma$ | $\theta_1$ | $\theta_2$ | $\varepsilon$ |
|---|---|---|---|---|---|---|---|---|---|
| 1. Reference group | 0.96, 0.02, 0.02 | 0.45 | 0.40 | 0.06 | 0.04 | 0.1 | 0.05 | 0.02 | 0.05 |
| 2. Experimental group 1 | 0.96, 0.02, 0.02 | 0.45 | 0.40 | 0.06 | 0.04 | 0.1 | 0.05 | 0.02 | 0.08 |
| 3. Experimental group 2 | 0.96, 0.02, 0.02 | 0.45 | 0.40 | 0.06 | 0.04 | 0.1 | 0.05 | 0.02 | 0.03 |

representing both negative ($In(t)$) and positive ($Ip(t)$) emotional disseminators decelerate in their descent. This implies that, in a state of equilibrium, there is a marked surge in both negative and positive emotional propagators. Furthermore, the $R(t)$ trajectory, indicative of the immune population within this digital domain, manifests a decline of nearly ten percentage points relative to the control scenario. This showcases that a sizable proportion of what should ideally be the immune populace in the equilibrium phase is, due to an elevated degradation rate, actively participating and propelling the event further. Inversely, a diminishing degradation rate fosters an upswing in the equilibrium-phase immune individuals and a commensurate downswing in emotional disseminators.

Interpreting this in the milieu of an epidemic, we discern that during periods of heightened epidemic-related crises or amidst escalating epidemic preventive pressures, individuals, once exposed to pertinent events, are predisposed to delve deeper into related news narratives, gravitating back towards susceptibility. This short-circuits the typical immune duration of the populace. A truncated average immunity span results in sustained high levels of emotion propagation within the network. To counteract this, under epidemic containment frameworks, governmental bodies ought to consistently enforce and communicate robust preventive measures. This would mitigate the overarching epidemic fervor, ensuring that societal emotional propagation remains subdued and stable. Consequently, this would attenuate the degradation rate for the digitally immune populace, extending the collective immunity duration across the online community.

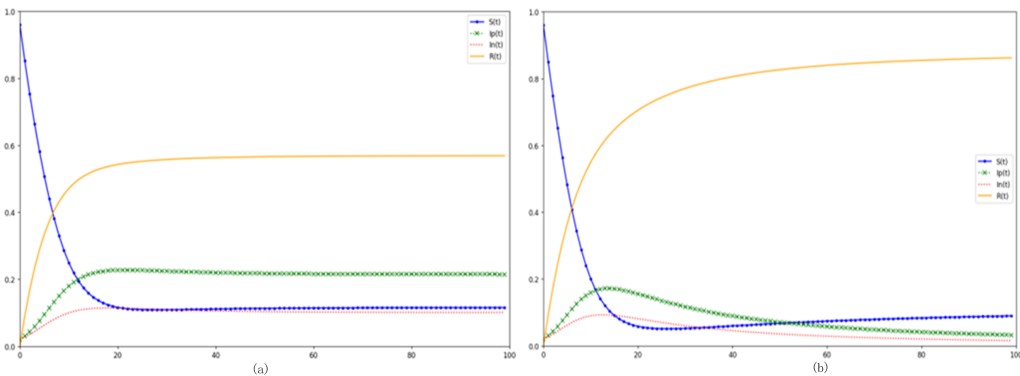

**Figure 6  Delving into the dynamics of degradation rate on digital emotional evolution.** (A) Increased degradation rate $\varepsilon = 0.08$. (B) Reduced degradation rate $\varepsilon = 0.03$.

## SIpInRS enhancement model grounded in complex networks
### Decoding the state transition probability function dynamics

In the given model, let's envision $g_i$ as a distinct node $i$ within the complex network, with its degree being symbolized by $k(g_i)$. The cohort of nodes adjacent to $g_i$ can be described as $\Gamma(g_i)$. Within this proximity, subsets of individuals, segmented by their emotional state—be it susceptible $\Gamma_S(g_i)$, positive emotion propagators $\Gamma_{Ip}(g_i)$, negative emotion propagators $\Gamma_{In}(g_i)$, or emotionally immune $\Gamma_R(g_i)$—are clearly delineated.

Now, if we represent the accumulated degrees of these neighboring nodes at a time $t$ as $d(t) = \sum_{g_m \in \Gamma(g_i)} k(g_m)$, we can further categorize them based on the emotional state of the nodes. Hence, $d_s(t)$, $d_{Ip}(t)$, $d_{In}(t)$, and $d_R(t)$ encapsulate the collective degrees of neighboring nodes corresponding to the susceptible, positive emotional propagators, negative emotional propagators, and emotionally immune nodes respectively, at that moment $t$.

Drawing from prior academic insights, during the progression of emotional contagion, an individual categorized as 'susceptible' is invariably confronted with both optimistic and pessimistic emotions. Intriguingly, these dual forces tend to counteract each other, leading to a neutralization of their cumulative impact (*Li et al., 2021*). Taking this phenomenon into account, if the collective degree of proximate positive emotion propagators $d_{Ip}(t)$ overtakes that of negative propagators $d_{In}(t)$, it's rational to infer that our reference node-currently deemed susceptible-stands a higher likelihood of evolving into a conduit of positive emotions. Contrarily, the influence of negative emotions gains precedence.

Factoring in the degree size of the node $g_i$ is pivotal. As a rule of thumb, nodes boasting a more substantial degree manifest a resilience against influences from adjacent nodes. Consequently, $Fs_{Ip}(g_i)$ embodies the probability of a susceptible node $i$ metamorphosing into a positive emotion propagator, while $Fs_{In}(g_i)$ epitomizes the likelihood of its transition into a purveyor of negative emotions. Our infection function is articulated, drawing

inspiration from preceding scholarly works.

$$A = |d_{Ip}(t) - d_{In}(t)| \tag{4}$$

$$F_{SR}(g_i) = \gamma \frac{d_S(t) + d_R(t)}{d_S(t) + d_R(t) + A} \tag{5}$$

$$F_{SIp}(g_i) = \alpha_1 \frac{2Ad_{Ip}(t)}{k(g_i)(d_{Ip}(t) + d_{In}(t))} \left(\text{when } d_{Ip}(t) > d_{In}(t)\right) \tag{6}$$

$$F_{SIn}(g_i) = \alpha_2 \frac{2Ad_{In}(t)}{k(g_i)(d_{Ip}(t) + d_{In}(t))} \left(\text{when } d_{Ip}(t) < d_{In}(t)\right) \tag{7}$$

It's imperative to note that a node predominantly disseminating negative emotions isn't immune to external influences. In fact, it remains susceptible to the sway of surrounding positive emotion propagators as well as individuals devoid of strong emotional hues (encompassing both immune and susceptible entities). Should the degree of a positive emotion propagator surpass its negative counterpart, there is a tangible probability of the negative node undergoing a 'purification', subsequently emerging as a beacon of positivity. The inverse scenario remains equally plausible. Thus, the probability metrics governing this transformative interplay between positive and negative propagators are meticulously defined.

$$F_{InIp}(g_i) = \theta_1 \frac{A}{k(g_i)} \left(\text{when } d_{Ip}(t) > d_{In}(t)\right) \tag{8}$$

$$F_{IpIn}(g_i) = \theta_2 \frac{A}{k(g_i)} \left(\text{when } d_{Ip}(t) < d_{In}(t)\right) \tag{9}$$

Furthermore, a heightened presence of emotionally neutral entities amplifies the susceptibility of emotional propagators to adopt an immune disposition. This transition probability towards immunization is also meticulously outlined.

$$F_{IpR}(g_i) = \beta_1 \frac{2(d_S(t) + d_R(t))}{d_S(t) + d_R(t) + A} \tag{10}$$

$$F_{InR}(g_i) = \beta_2 \frac{2(d_S(t) + d_R(t))}{d_S(t) + d_R(t) + A} \tag{11}$$

Leveraging these defined probabilities, we have successfully sculpted a refined version of the SIpInRS model.

$$\frac{dS(t)}{dt} = -F_{SIp}(g_i)S(t)Ip(t) - F_{SIn}(g_i)S(t)In(t) - F_{SR}(g_i)S(t) + \varepsilon R(t) \tag{12}$$

$$\frac{dIp}{dt} = F_{SIp}(g_i)S(t)Ip(t) + F_{InIp}(g_i)In(t) - F_{IpIn}(g_i)Ip(t) - F_{IpR}(g_i)Ip(t) \tag{13}$$

$$\frac{dIn}{dt} = F_{SIn}(g_i)S(t)In(t) + F_{IpIn}(g_i)Ip(t) - F_{InIp}(g_i)In(t) - F_{InR}(g_i)In(t) \tag{14}$$

$$\frac{dR}{dt} = F_{IpR}(g_i)Ip(t) + F_{InR}(g_i)In(t) + F_{SR}(g_i)S(t) - \varepsilon R(t) \tag{15}$$

**Table 6 Parameters of the network.**

| Network | Number of nodes | Edge number | Average degree | Degree correlation coefficient | Clustering coefficient | Maximum nodal degree |
|---|---|---|---|---|---|---|
| 1. BA scale-free network | 10,000 | 39,998 | 5.32 | −0.032 | 0.018 | 347 |
| 2. WS Small world Network | 10,000 | 40,000 | 8.01 | −0.041 | 0.126 | 15 |
| 3. Sina Weibo | 31,325 | 53,264 | 3.13 | −0.328 | 0.054 | 2,000 |

## Analysis of model simulation

By employing relevant algorithms, Watts–Strogatz's (WS) small-world network, Barab'asi-Albert's (BA) scale-free network were crafted. Subsequent simulations were carried out using real-life network datasets from microblogs. The intricate topological data pertinent to network parameters is presented in Table 6.

(1) Insight into the network structure's impact on emotional contagion

To maintain consistency, the experiments were initialized based on the reference group parameters established in the kinetic model. Breakdown of initial proportions was: Susceptible at 0.96, Positive Emotion Propagators at 0.02, and Negative Emotion Propagators at 0.02.

These visuals paint a comprehensive picture of state node proportions as contagion rounds evolve (as shown in Fig. 7). Across the three models, the $x$-axis represents contagion rounds, while the $y$-axis details the tally of respective state nodes. Interestingly, the contagion pattern largely mirrors the latent four-phase lifecycle model. The degradation rate exerts an influence - after an initial rapid susceptibility downturn during the outbreak phase, a subsequent gradual rise occurs owing to an increasing cohort of immune individuals. This process culminates in a stable equilibrium.

Figures 7A–7C show the trends of different state node proportions with the development of contagion rounds in the BA scale-free network, WS small-world network, and SIpInRS improved model of the Sina Weibo network, respectively.

A striking observation from the models reveals that the WS small-world network reaches its emotional propagation peak around $t = 5$. Conversely, the BA scale-free network hits its zenith earlier, with the negative emotion curve peaking at $t = 3$. Notably, this model's ascent both in terms of speed during the initial burst and peak magnitude exceeds the WS small-world network. This resonates with prior scholarly insights suggesting that contagion velocities in non-uniform networks overshadow their uniform counterparts (*Ran & Chen, 2021*).

Contrastingly, the real-world Sina Weibo network's contagion pace is more gradual, and its peak, subdued when juxtaposed against the BA network. As illustrated in Fig. 8, the unique topology of the Sina Weibo network offers an explanation. Despite interconnected subclusters, inter-cluster connections are sparse, hindering seamless information flow. Another distinct attribute is Sina Weibo's follower cap at 2000, fostering a pronounced presence of "super disseminators". This dynamic is starkly different from platforms like Twitter, which yields a pronounced cluster phenomenon. Given the pivotal role of super disseminators in emotional contagion, coupled with microblogs' intrinsic influence

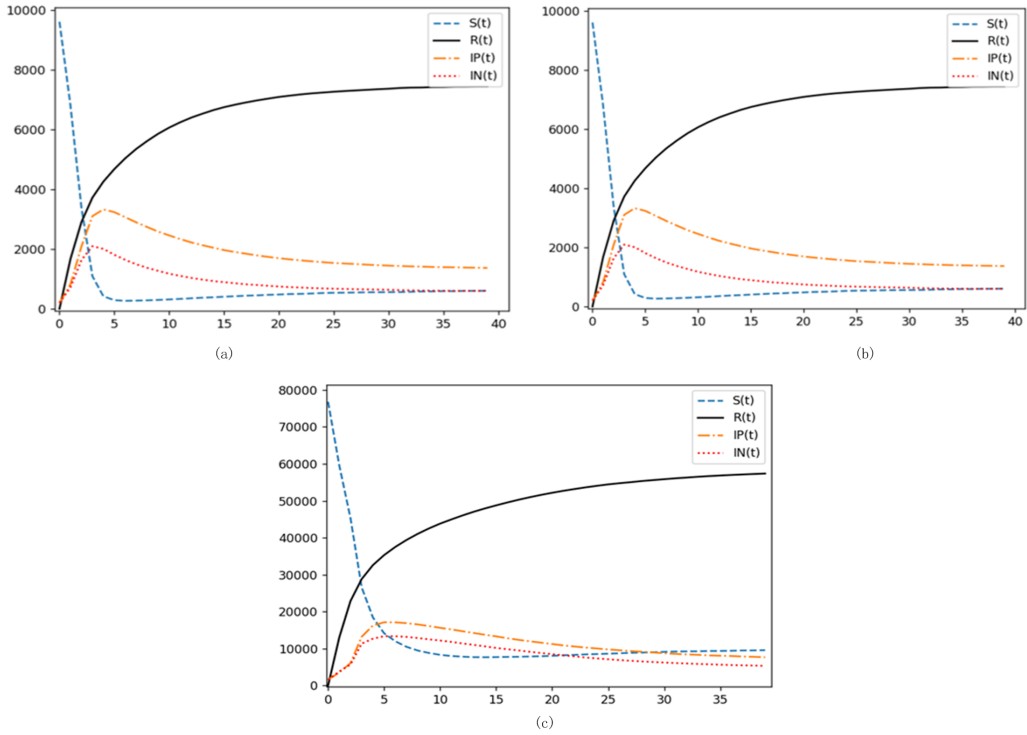

**Figure 7** **Network structural impact on emotional contagion.** (A) BA scale-free network simulation results. (B) WS small-world network simulation results. (C) Sina Weibo simulation results.

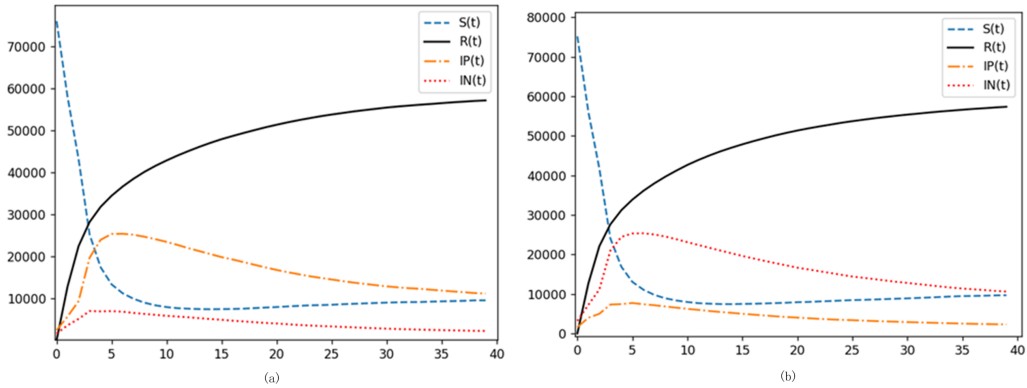

**Figure 8** **Examining the impact of initial disseminators on emotional contagion.** (A) Increase in the proportion of initial positive emotion transmitters. (B) Increase in the proportion of initial negative emotion transmitters.

on information dissemination, this stark clustering translates into reduced propagation efficiency and limited reach when paralleled with BA networks.

(2) Analysis of the influence of initial disseminators' proportion on emotional spread

Our primary objective is to unravel the intricacies of contagion within actual real-world networks. Thus, taking the Sina Weibo network delineated in Fig. 8 as a baseline, we

expanded the nodes representing initial positive disseminators and those signifying initial negative emotion propagators. This exercise was aimed at understanding the interplay of these factors on emotional contagion within a multifaceted network landscape.

As is evident from Figs. 8A and 8B, simulation outcomes bore a striking resemblance to trends observed under the kinetic model. Augmenting the proportion of initial disseminators compresses the duration of the emotional outburst phase while amplifying the intensity of the peak. This corroborates the assertions made in existing literature (*Xiong et al., 2018*). An intriguing distinction emerges: in the kinetic model, bolstering the proportion of one emotional disseminator doesn't markedly skew the opposite emotion. However, in real-world network topologies, the tug of war between positive and negative emotions is accentuated. As one emotional node gains traction, it simultaneously quells the opposite emotion. Using negativity as our lens, we discern that a proliferation of nodes radiating negativity not only exposes more nodes to this emotion but also sways nodes originally aligned with positivity. This cyclical amplification, where negativity spawns more negativity, throws the network off-kilter during the outbreak. Eventually, when emotions plateau, negative nodes still hold a numerical advantage.

This dynamic captures a quintessential real-world phenomenon: post unexpected events, online emotional currents often flow unidirectionally.

Drawing from this elucidation, a salient takeaway for policymakers emerges: the formative stages of emotional spread warrant keen oversight. Negligence here can usher in a domino effect, with negativity reigning supreme during event outbursts. Thus, a proactive approach during the latency period can potentially avert the spiraling negativity, mitigating its overarching dominance during event eruptions.

(3) Analysis of the impact of initial disseminator node degree on emotional contagion

Digital ecosystems, especially social networks, are heterogeneous in nature. Influence varies dramatically across users, with certain nodes, such as opinion leaders, becoming pivotal in steering sentiment. A case in point is the Weibo vlogger, whose sway can outweigh that of ordinary users by magnitudes. The node degree serves as a proxy for such influence. The potency of the initial contagion node's degree stands paramount in sentiment propagation.

While Fig. 9 illustrates the sentiment spread emanating from a randomly chosen initial node, a more nuanced approach is adopted subsequently. By arranging nodes based on their degree magnitude, Figs. 9A and 9B delineate scenarios where the initial positive and negative emotion propagators, respectively, are nodes with substantial degrees. Echoing *Zhang, Feng & Yang (2019)*, a greater node degree can turbocharge emotional contagion.

Simulations reveal a thematic consistency: the influence of the initial propagator's node degree on emotion contagion parallels the impact of bolstering node count. A surge in the corresponding emotion's propagation velocity is observed, accompanied by an earlier and heightened peak, in line with findings by *Wei et al. (2021)*. Crucially, the node degree plays an instrumental role in sentiment spread, potentially even surpassing the effects of augmenting the initial node proportion, especially in neutralizing contrary emotions.

In real-world digital landscapes, opinion leaders command outsized influence. Even in the face of dissenting views, the majority, swayed by the herd mentality, often gravitates

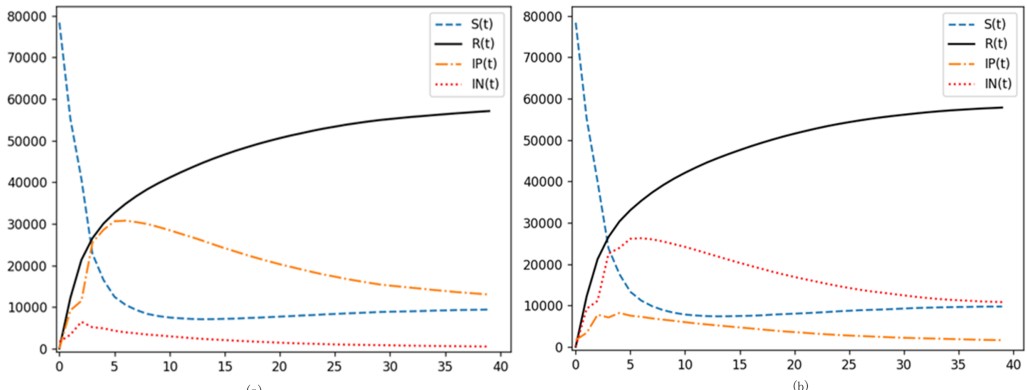

**Figure 9** **Evaluating the influence of initial disseminator node degree on emotional spread.** (A) Increasing the initial positive emotion transmitter node degree. (B) Increasing the initial negative emotion propagator node size.

towards these influencers' perspectives. This exemplifies the formidable, and sometimes daunting, leverage of opinion leaders in the digital emotional arena. Consequently, prominent internet personas, especially high-degree nodes, must exercise prudence. Their substantial reach demands responsibility. By setting positive precedents and disseminating objective, balanced sentiments, especially at the onset of events, they can shepherd online emotional currents towards constructive trajectories.

(4) Analysis of the degradation rate's impact on emotional contagion

The unique trajectory of the novel coronavirus, characterized by recurring flare-ups, renders it an unyielding challenge. Even during moments of relative tranquility, public concern remains palpable, amplifying during heightened outbreak phases. The pervasive and recurrent nature of the epidemic sentiment poses unique challenges for control measures. Reflecting on global policy responses, a majority of nations have opted for a strategy of virus coexistence, suggesting that the current dynamic may persist indefinitely. Hence, the SIRS base model, augmented with a degradation rate, provides a more apt framework for investigating emotional contagion linked to epidemic narratives.

Figure 10 elucidates the kinetic equation's nuanced interplay with degradation rate. The degradation rate, which dictates the likelihood of the immune reverting to susceptibility and subsequently becoming emotion disseminators, primarily shapes the latter stages of the emotional contagion cycle. As degradation rates rise, emotions wane more gradually, culminating in a diminished immune proportion when equilibrium ensues. Within real-world digital networks, the degradation rate is intrinsically tied to epidemic trajectory and preventive measures. A stable recent epidemic history, coupled with adept control mechanisms, tends to lessen public fixation on the crisis. Consequently, this reduces the degradation rate, as individuals gradually shift focus away from the epidemic.

## Comparative analysis with actual data

Data from Sina Weibo, pertaining to the epidemic in Sanya, Hainan, from August 2022, were extracted and analyzed. This real-world data was juxtaposed against our simulation

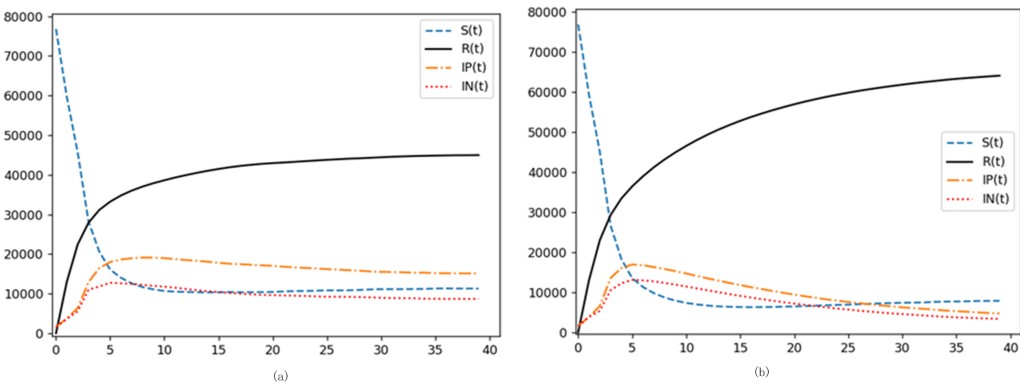

**Figure 10** **Probing the impact of degradation rate on emotional contagion.** (A) Increased degradation rate. (B) Reduced degradation rate.

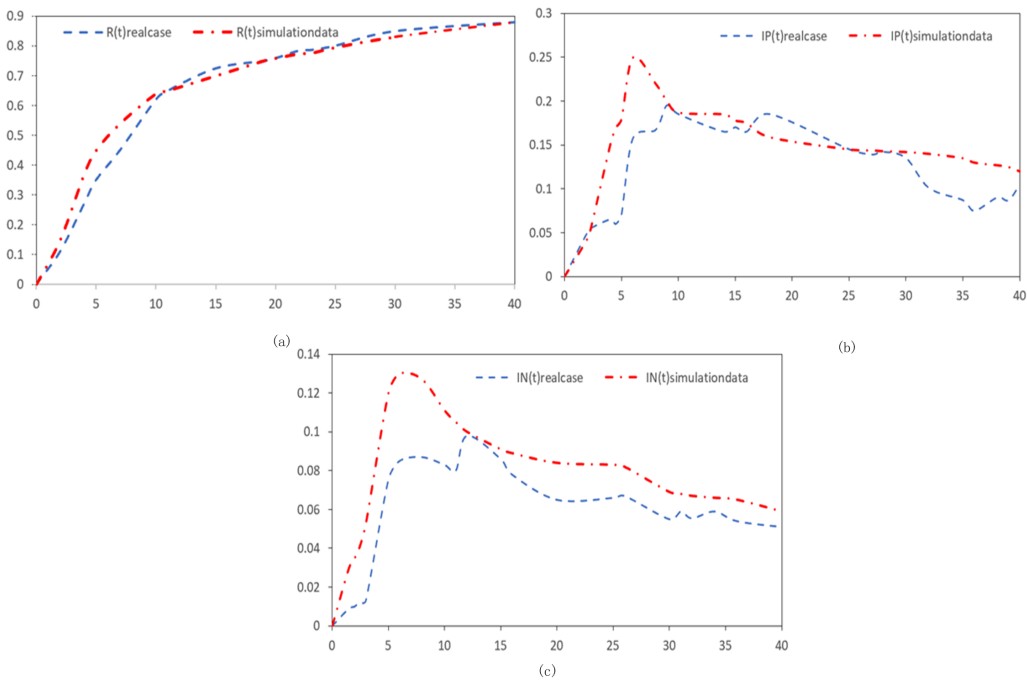

**Figure 11** **Real data synthesis.** (A) Comparison of curve fitting effect of immune subjects (B) Comparison of curve fitting effects of positive emotion disseminators. (C) Comparison of curve fitting effects of negative emotion disseminators.

outcomes. The initial confirmed case on August 1 was designated as the reference point, $t_0$. The daily ratio of positive to negative sentiment disseminators was derived from the total count of corresponding sentiment-laden tweets, divided by the day's overall tweet volume. The model incorporated parameters inspired by previous control group findings. The $x$-axis delineates the temporal evolution, while the $y$-axis captures the overarching proportion for each emotional state. These comparative findings are illustrated in Fig. 11.

The comparative insights reveal a remarkable congruence between actual data trends and our simulation projections. Notably, our model offers a more nuanced and faithful representation of netizen sentiment dynamics during the novel coronavirus epidemic when contrasted with the conventional SISR framework. This underscores the model's potential in simulating real-world scenarios and forecasting netizen reactions amidst health crises.

## CONCLUSIONS

Building upon traditional infectious disease models, an SIpInRS model for emotion contagion amongst netizens was devised, incorporating elements that influence emotion contagion within a genuine epidemic context. The degradation rate, reflecting the unique nature of the novel coronavirus epidemic, was introduced, elevating the model's adherence to contagion dynamics prevalent during this pandemic. Furthermore, the classification of emotion disseminators into positive and negative categories brought additional granularity, further aligning the model with observed epidemic sentiment patterns.

Simulation experiments illuminated several pivotal dynamics:

- The infection rate was found to influence the velocity of emotion spread.
- Both incitement and purification rates were identified as determinants shaping the overarching emotional orientation of the network.
- The immunity rate was discerned to influence the proportional distribution across emotional states at equilibrium.

The SIpInRS model underwent further refinement, entailing the definition of a contagion probability function interlinked with the inherent topology of complex networks. Simulated outcomes revealed that an increased count of initial emotion disseminators, coupled with an augmented initial contagion node degree, can potentiate the rapidity and peak amplitude of emotion contagion across the broader social network matrix. Moreover, an upsurge in either positive or negative emotions was observed to markedly suppress its counterpart. Pertinently, the degradation rate emerged as a crucial factor, impacting the deceleration of emotion contagion in intermediary and subsequent phases, and dictating the terminal equilibrium proportions of various emotional states.

### Funding

This work is supported by the National Natural Science Foundation of China (Grant number: 72061147005; 71874215; 72004244; 71571191), the National Social Science Foundation of China (Grant number: 21BZZ108), the Beijing Natural Science Foundation (Grant number: 9182016; 9194031); the MOE (Ministry of Education in China) Project of Humanities and Social Sciences (Grant number: 17YJAZH120; 19YJCZH253), the Fundamental Research Funds for the Central Universities (SKZZY2015021) and the Political Education Special Fund in Central University of Finance and Economics ("Diffusion mechanism and synergistic analysis of epidemic prevention policies during

COVID-19 epidemics", SZJ2208). The funders had no role in study design, data collection and analysis, decision to publish, or preparation of the manuscript.

## Grant Disclosures

The following grant information was disclosed by the authors:

National Natural Science Foundation of China: 72061147005, 71874215, 72004244, 71571191.

National Social Science Foundation of China: 21BZZ108.

Beijing Natural Science Foundation: 9182016, 9194031.

MOE: 17YJAZH120, 19YJCZH253.

Fundamental Research Funds for the Central Universities: SKZZY2015021.

Political Education Special Fund in Central University of Finance and Economics: SZJ2208.

## Competing Interests

The authors declare there are no competing interests.

## Author Contributions

- Yanchun Zhu analyzed the data, performed the computation work, authored or reviewed drafts of the article, and approved the final draft.
- Wei Zhang performed the experiments, performed the computation work, prepared figures and/or tables, and approved the final draft.
- Chenguang Li conceived and designed the experiments, authored or reviewed drafts of the article, and approved the final draft.

## Data Availability

The raw data and code are available in the Supplemental Files.

## Supplemental Information

Supplemental information for this article can be found online at http://dx.doi.org/10.7717/peerj-cs.1693#supplemental-information.

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
