# Peer review of "Modeling emotional contagion in the COVID-19 pandemic: a complex network approach"

_PeerJ Computer Science, doi:10.7717/peerj-cs.1693_

## Round 0.1 · original submission · Major Revisions

This work is interesting and has value. However, the current version also has issues needing improvement both in writing and technical aspects. Please find the detailed comments from reviewers and revise the paper accordingly. Then it will be evaluated again,

**Language Note:** The review process has identified that the English language must be improved. PeerJ can provide language editing services - please contact us at copyediting@peerj.com for pricing (be sure to provide your manuscript number and title). Alternatively, you should make your own arrangements to improve the language quality and provide details in your response letter. – PeerJ Staff

Reviewer 1 ·

Basic reporting

English editing is absolutely needed as some grammar mistakes exist throughout the manuscript.

Some terms and abbreviations including acronyms are missing (e.g., WS and BA ind the Abstract).

The introduction and literature review are focused at computational modeling but are missing the aspect of psychology for describing modeling of human behavior in regards to negative and positive feelings.

Starting at the Abstract and throughout the manuscript, it is not always clear if the authors describe biological contagion or emotional contagion. Please make this clear.

The motivation for using biological infection models to describe human behavior is not intuitive and should be better explained in the Abstract and the Introduction. Existing models have adopted biological models to describe purchase discussions, and online engagement (e.g. re-share information) but indicate that biological models fail to correctly describe human behavior.

The 1st sentence in the Abstract is 3 lines long and not clear. Please make it shorter. 4

Line 64: the authors mention "real cases" how ever, it is not clear what are these real cases showing.

Line 25, why is negative a referred to as a spreader and positive is referred to as a communicator?

Experimental design

The authors do not indicate why (according to their claims) people are getting angry.

In addition, the authors did not define what is considered a negative or positive emotion.

The authors discuss individual differences but it is not clear what are those differences. Are they emotional or other differences (e.g., age).

In line 120, the authors mention "Energy". What is positive energy? how would you measure optimistic level? it is not clear and not well defined in the manuscript.

The authors give examples to negative emotions such as falsehood, panic and suspicion. I disagree that suspicion is a negative emotion.

line 150: gamma indicates that infection depends on the person and how much susceptible s/he is.
Human behavior in adopting a behavior or a product is much more complex that a single variable (see Laato, S., Islam, A. N., Farooq, A., & Dhir, A. (2020). Unusual purchasing behavior during the early stages of the COVID-19 pandemic: The stimulus-organism-response approach. Journal of Retailing and Consumer Services, 57, 10222).

Continuing the last point discussing complex behavior, what happens if a person has mixed feelings? Can't s/he be infected with positive and negative feelings?
This is very similar to a topic model where a document can affiliate with more that one topic.

Validity of the findings

I find it very difficult to assess the validity of the reported results as they are mostly based on simulations.
The authors do present a single case of Weibo data.
It is not clear what is the 1st confirmed case? is it an infection of a biological disease or a human feeling infection?
In addition, the authors do not compare their model to other models to support their results.

Indeed, the curves in Figure 11 show similarity of simulation to real world data. However, it is not clear whether this similarity is significant. I suggest reporting the outcome of a statistical test.

Additional comments

The dataset and code are not available.

Cite this review as

Reviewer 2 ·

Basic reporting

Exploring emotion propagation characteristics during public health emergencies is an important topic which plays a major role in optimizing epidemic policies and enhancing prevention behavioral compliance, and etc. In emotion propagation, positive and negative emotions coexist and propagate in a competing manner across complex networks of people, and thus make emotion propagation analysis more challenging. By considering the emotion propagation process and individual connectivity, the authors explore the effect of several parameters on emotional propagation, e.g., emotional infection rate, initial propagation ratio, degradation rate, initial communicator node degree, and etc.

The study can provide guidelines for the applications of emotion propagation. The topic is relatively new, and the problem is of significance. However, there are several flaws in the paper that compromise the contributions of this paper:
1. Authors analyze the process of competitive propagation of positive and negative emotions. However, they do not consider the interaction of epidemic spread with emotion propagation but emotion propagation alone. This is not in line with the paper that proposes “Existing work lacked consideration of the emotion interaction in the epidemic scenario.”. So, it will be more persuasive to take into account the interaction between emotions and epidemics.
2. In related research, it is necessary to summarize the related work and present the inheritance and innovation of your own work in relation to them at the end.
3. In Section SIpInRS improvement model based on complex networks, authors define the $\Gamma_S(g_i), \Gamma_{Ip}(g_i), \Gamma_{In}(g_i) $, and $\Gamma_{R}(g_i)$, but do not use. Appropriate omissions are beneficial.
4. Authors devote too much segments to discuss the performance in the mean field. This is not relevant to “A Complex Network Perspective” in the title.
5. Authors define $F_{SIp}(g_i)=\alpha_1 \frac{2 A d_{Ip}(t)}{k(g_i) (d_{Ip}(t) + d_{In}(t))}$ as the probability that susceptible node i transforms into a positive emotion transmitter. It is necessary to explain the physical meaning and derivation of the equation.

Experimental design

1. In model simulation analysis, it is necessary to indicate the specific parameters of each experiment.
2. Almost all the experimental images in the paper are blurry, especially Fig. 11(a). It is necessary to use high-definition experimental images.

Validity of the findings

no comment

Additional comments

In addition, there are some typos in the paper. The authors may want to check them out to improve the presentation. For example,
(1) Throughout the paper, the words propagation, transmission, and contagion are mixed. It is a good habit to use them in a uniform way.
(2) In Section model equilibrium point solution and stability analysis, “Therefore, for the zero equilibrium point underP0” should be “Therefore, for the zero equilibrium point under P0”.
(3) In Section model equilibrium point solution and stability analysis, “the immunization rate of the infected person $\beta$ effectively controlling the corresponding transmission threshold while reducing the degradation rate $\varepsilon$” should be “the immunization rate of the infected person $\beta$ effectively controlling the corresponding transmission threshold while reducing the degradation rate $\varepsilon$.”.
(4) In Section model equilibrium point solution and stability analysis, the sentence, “On the other hand, we can also use self-media to guide people's thoughts positively so that infected people can be transformed into immune people faster and thus achieve effective control of the overall emotion of the internet.”, only has one “on the other hand”, you should add “on the one hand” or delete it.
(5) Some equations are not centered, e.g., Equation (4), (8), and (9).

Cite this review as

---

## Round 0.2 · accepted · Accept

Thanks to the authors for your efforts to improve the work.

Reviewer 2 ·

Basic reporting

The author has addressed all my concerns, and I have no more comments

Experimental design

No comment

Validity of the findings

No comment

Additional comments

No comment

Cite this review as